**Data Availability Statement:** All relevant data are within the paper and its Supporting information files.

**Funding:** YES. Sponsored by Fujian Provincial Science and Technology Department Social

# Multi-angle laser device improves novice learning of C-arm fluoroscopy for lumbar spine surgery

Yuan-Dong Zhuang[1☯], Rui-Jin Li[2☯], Jia-Jun Wu[2], Xue-Wei He[2], Wen-Bin Zou[2], Xu-Chu Xu[2], Si-Qi Lu[2], Chun-Mei Chen[1]*

1 Department of Neurosurgery, Fujian Institute of Neurosurgery, Fujian Medical University Union Hospital, Gulou District, Fuzhou, Fujian, China, 2 Fujian Medical University, Minhou County, Fuzhou, Fujian, China

☯ These authors contributed equally to this work.
* cmchen2009@sina.com

## Abstract

### Purpose

This study aims to evaluate the efficacy and satisfaction of using a multi-angle laser device (MLD) for C-arm fluoroscopy to assist novice learners during lumbar spine surgery.

### Methods

Forty novice learners were randomly assigned to Group A using an MLD-equipped C-arm or Group B using a traditional C-arm. Both groups performed X-ray fluoroscopy on a lumbar spine model in supine and rotated positions. Time, number of shots, and deviation from the target were compared. A questionnaire was used to assess the learning experience.

### Results

Group A required less time (13.66 vs. 25.63 min), and fewer shots (15.05 vs. 32.50), and had a smaller deviation (22.9% vs. 61.5%) than Group B (all p<0.05). The questionnaire revealed higher scores in Group A for comfort, efficiency, and knowledge mastery (all p<0.05).

### Conclusion

The MLD significantly improves novice learning of C-arm fluoroscopy during lumbar spine surgery.

## 1 Introduction

C-arm fluoroscopy is critical for precise surgical localization during minimally invasive lumbar spine surgery [1]. However, novice operators often require prolonged radiation exposure due

development guidance (key) project(Grant number: 2020Y0034 to ZHUANG Yuan-Dong) - contributed to study design and data collection Fujian Provincial Joint Funds for the Innovation of Science and Technology(Grant number: 2021Y9061 to ZHUANG Yuan-Dong) - contributed to data analysis Ministry of Education Industry-school Cooperative Education Project (Grant number: 220602999165419 to ZHUANG Yuan-Dong) - contributed to decision to publish United Fujian Provincial Health and Education Project for Tackling the Key Research. P.R. China (Grant number: 2019-WJ-08 to CHEN Chun-Mei) - contributed to preparation of the manuscript.

**Competing interests:** The authors have declared that no competing interests exist.

**Abbreviations:** AP, Anteroposterior; CFD, CMOS flat detector (GE HealthCare); CR, coronally rotated position; LAT, Lateral; MLD, Multi-angle laser device; N, Normal prone position.

to lack of experience, which increases risks of surgical complications [2]. This not only increases surgical time and radiation exposure but also impedes learning efficiency [3,4].

Lumbar spinal surgery can risk nerve injury, infections, hemorrhage and other complications [Zhuang, 2023 #3456] [ZHUANG, 2017 #2349] [5], especially among older patients with comorbidities [Zhuang, 2024 #3457] [6]. Surgeries involving sacrum fusion also increase risks compared to short segment procedures [7]. Therefore, developing precise and efficient C-arm imaging technique is critical for novice learners to minimize localization errors and reduce complication rates.

To address this issue, our team developed a multi-angle laser device (MLD) for C-arm machines. The MLD establishes a spatial correspondence between lasers, X-rays, and the C-arm to achieve precise localization on the patient's body surface, enabling faster acquisition of accurate intraoperative images. This technology aims to reduce fluoroscopic exposures, enhance imaging accuracy, minimize radiation dosage, and assist novice operators in learning C-arm fluoroscopy techniques, thereby reducing the learning curve.

The objective of this study was to evaluate the efficacy and satisfaction of using the MLD to improve the efficiency of C-arm fluoroscopy and the learning experience among novice operators during simulated lumbar spine surgery. We hypothesized that the MLD would facilitate precise and efficient image acquisition while enhancing the comfort and knowledge mastery of novice learners.

## 2 Data and methods

### 2.1 Ethics statement

This study was approved by the Ethics Committee of Fujian Medical University Union Hospital (NO.2020KJT073). The need for informed consent was waived by the ethics committee, as the data were analyzed anonymously and no interventions were performed on the participants that could cause harm. 3D-printed human body models were utilized for the surgical simusectionlations, ensuring the safety of patients and students without causing any physical harm. The study was conducted in accordance with ethical principles for human subjects research, minimizing risk and protecting the welfare of the participants.

### 2.2 Participant selection

Forty second-year medical students without prior C-arm experience were recruited as novice operators in July 2022. They were enrolled in clinical medicine at the School of Basic Medical Sciences, Fujian Medical University. All participants had basic anatomy knowledge.

Participants were randomly assigned to Group A (n = 20) or Group B (n = 20). The groups were matched by gender ratio and baseline anatomy test scores to ensure homogeneity in the sample.

### 2.3 Experimental setup

**2.3.1 Environmental and equipment setup.** The study was conducted in the operating room at Fujian Medical University Union Hospital under consistent environmental conditions. The C-arm machine was equipped with the MLD and installed on an OEC One CFD (CMOS flat detector) mobile X-ray system (GE HealthCare Inc.). The MLD comprised perpendicular green line lasers at the X-ray tube and image intensifier, along with green and red cross lasers at 90˚, 45˚, and 135˚ angles (Fig 1).

**2.3.2 Phantom preparation.** A 3D-printed lumbar spine phantom was used to simulate the human lumbar region [Zhuang, 2019 #3156]. A needle was inserted into the right L4-L5

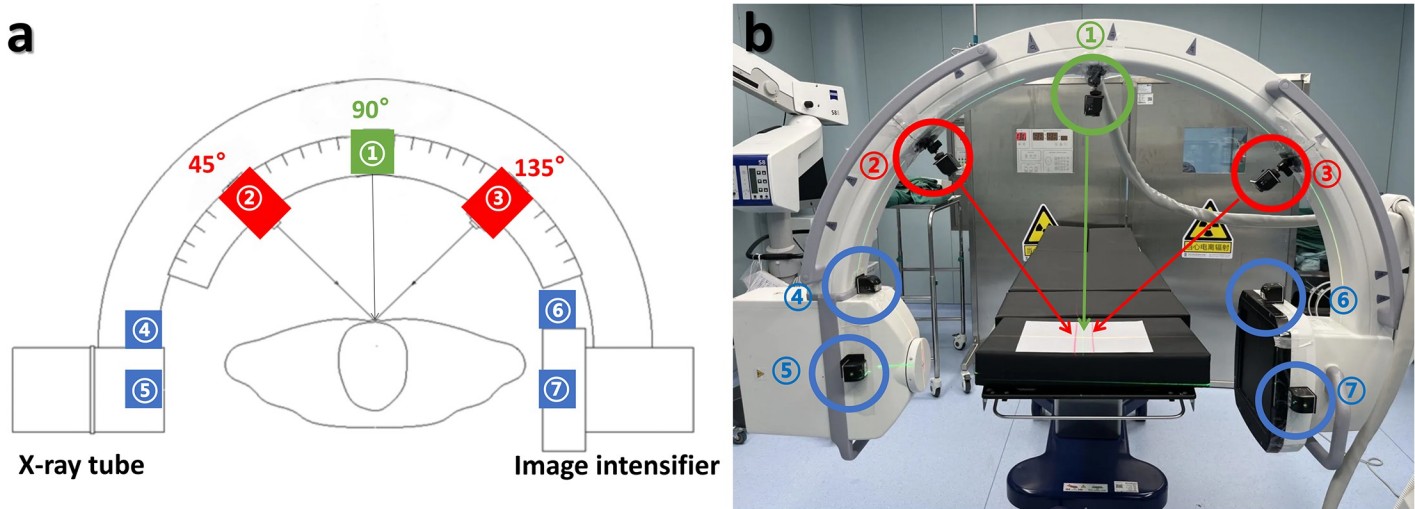

**Fig 1. Multi-angle laser device (MLD) was installed on the C-arm machine.** (a) Schematic diagram: The green cross laser (①) is positioned at the top of the C-arm at 90˚, and the red cross lasers (②③) are positioned at 45˚ and 135˚ on the C-arm. The green line lasers (④-⑦) are placed perpendicularly and positioned at the X-ray tube and image intensifier sides. (b) Photograph: The green circle corresponds to laser ①, the red circle corresponds to lasers ②③, and the blue circle corresponds to lasers ④-⑦.

intervertebral foramen to serve as the imaging target. The phantom was placed inside an opaque box on the surgical table in a standard prone position and a 45˚ coronally rotated position (Fig 2).

## 2.4 Two different imaging methods

Participants were divided into Group A using MLD-equipped C-arm and Group B using a traditional C-arm. Both groups performed fluoroscopy on the phantom in supine and 45˚ rotated positions to acquire anteroposterior (AP) and lateral (LAT) images (Fig 3). Group A utilized

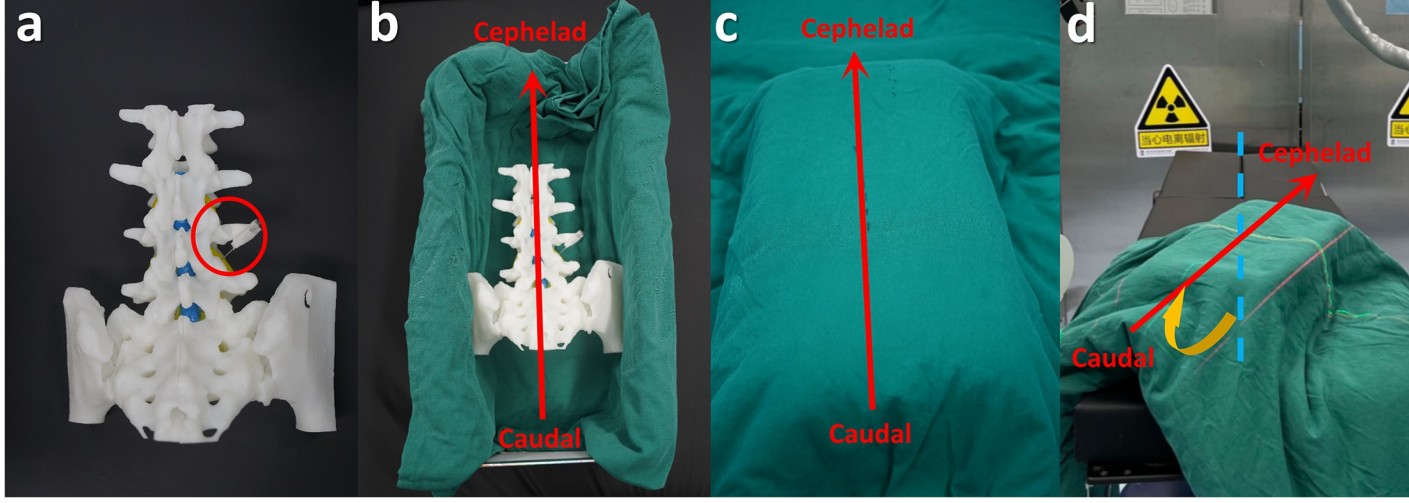

**Fig 2. Positioning of the lumbar spine phantom for imaging.** (a) 3D-printed phantom with an inserted needle as the imaging target. (b) The phantom was placed inside an opaque box on a surgical table. (c) Standard prone position. (d) 45˚ coronally rotated position.

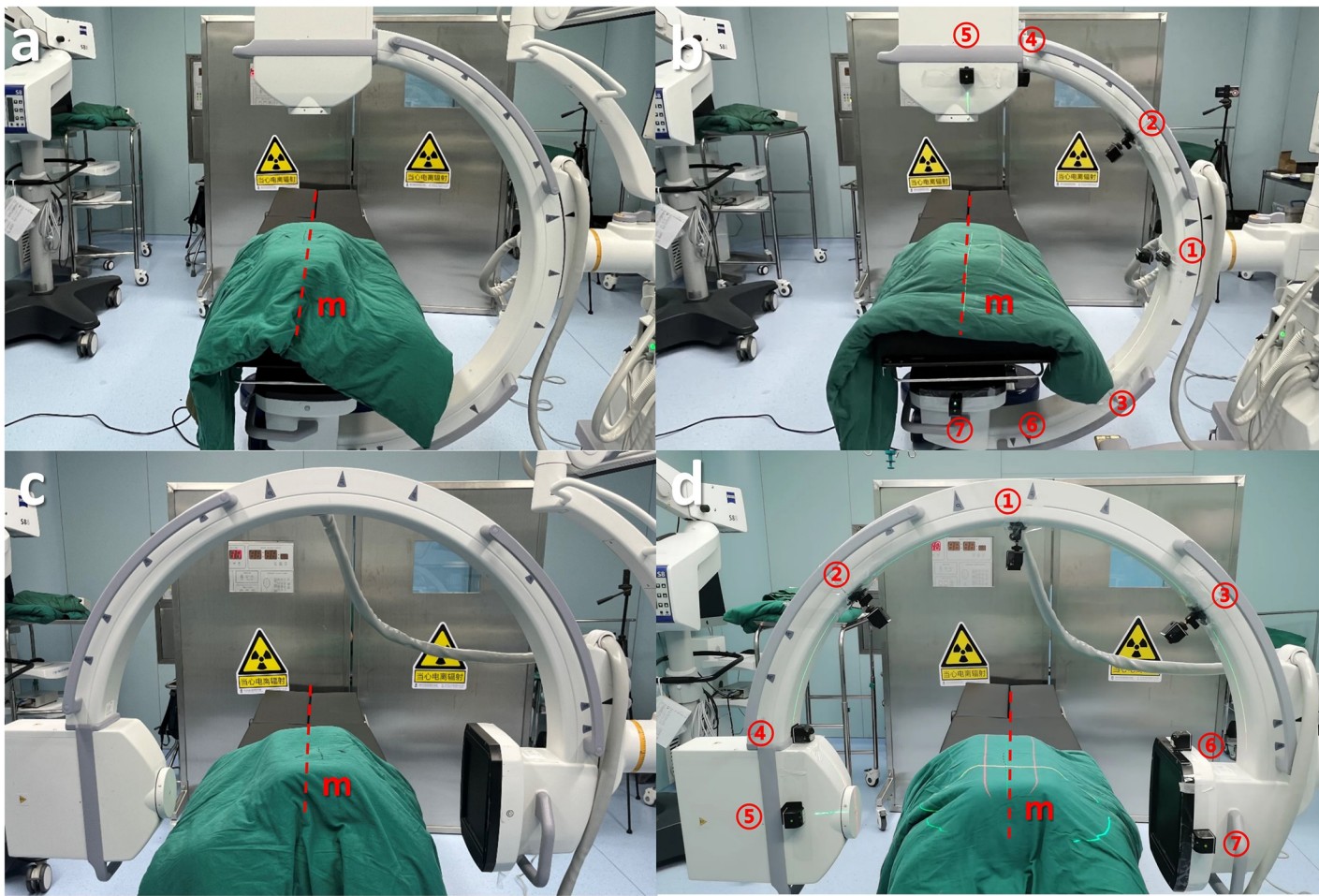

**Fig 3. Comparison of traditional versus multi-angle laser assisted imaging.** (a) Traditional AP imaging. (b) AP imaging with MLD guidance. (c) Traditional LAT imaging. (d) LAT imaging with MLD guidance. Note: MLD lasers helped align the X-ray beam perpendicular to the phantom's long axis (dashed line) before imaging.

perpendicular and angled lasers to determine the phantom orientation and align the C-arm's X-ray beam perpendicular to the long axis before imaging. Group B relied on repeated adjustments without laser guidance to obtain adequately aligned images.

## 2.5 Imaging endpoint criteria

The imaging endpoint was defined as capturing images with the needle tip target at the center. On the display screen, the center point was determined by the intersection of diagonal lines across the image (Fig 4a).

Images were captured repeatedly until the deviation of the needle tip from the center was less than 10% of the image radius (Fig 4b). The deviation was calculated as the distance between the needle tip and the image center divided by the radius.

Time, number of shots, and deviation were recorded for analysis.

## 2.6 Questionnaire

After imaging, a 5-point Likert scale questionnaire (1 = completely disagree, 5 = completely agree) was administered to assess four aspects of the learning experience:

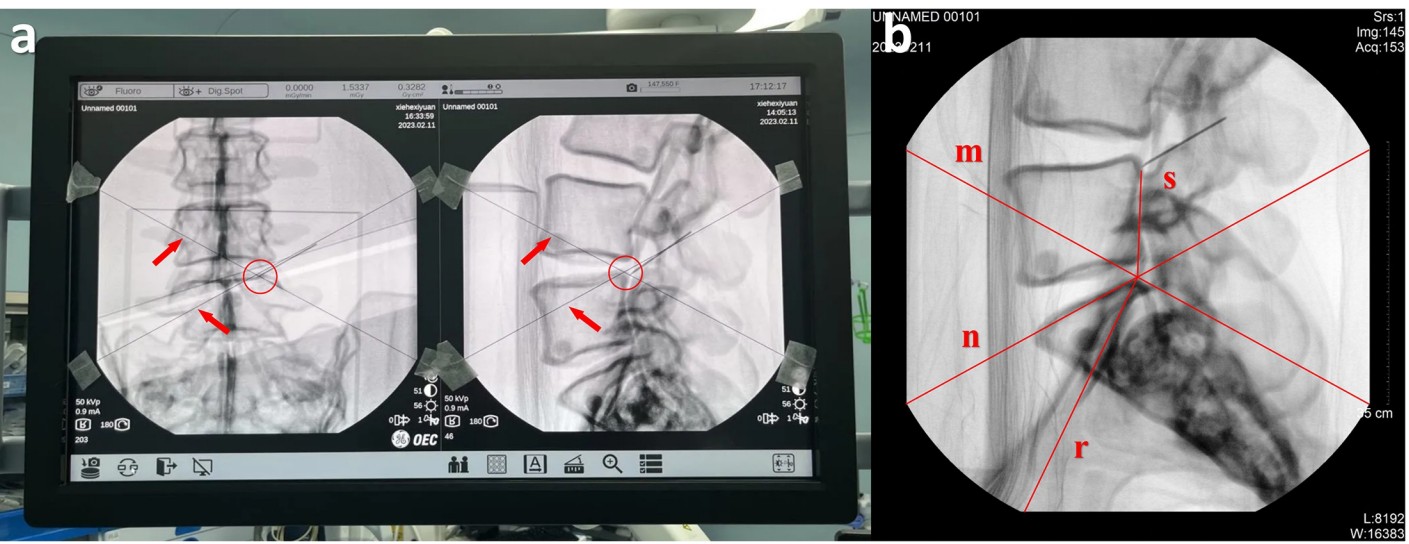

**Fig 4. Determination of the imaging endpoint.** (a) Center point identification. The intersection of diagonals indicates the image center. (b) Deviation measurement. Deviation percentage = Distance between the needle tip and center (s) / Image radius (r). Imaging ends when deviation <10%.

Comfort during imaging, Interest level, Self-learning efficiency, Mastery of procedural knowledge. The results were compared between Group A and Group B to evaluate the effectiveness and satisfaction of using the MLD-equipped C-arm.

### 2.7 Statistical methods

SPSS 25.0 was used for data analysis. The chi-square test was used to compare gender ratios. Descriptive statistics were presented as mean ± standard deviation. For normally distributed data, an independent t-test was used to compare between groups. Non-normally distributed data was compared by the Mann-Whitney U test. Statistical significance was defined as $p < 0.05$.

## 3 Results

### 3.1 Baseline consistency analysis of participants

There were no significant differences in gender ratio or anatomy scores between Group A and Group B (both p>0.05, Table 1). All participants had no prior C-arm experience.

### 3.2 Analysis of participants' operational data

The operational data shows Group A required less time (mean 13.66±5.61 min) than Group B (25.63±8.17 min), a 46.7% reduction. Group A also had fewer shots (mean 15.05±7.84) than Group B (32.50±7.24), a 53.7% decrease. Additionally, the initial deviation was smaller in

**Table 1. Comparison of baseline characteristics.**

| Item | Group A(n = 20) | Group B(n = 20) | P-value |
|---|---|---|---|
| Male | 13 | 14 | 0.736(P>0.05) |
| Female | 7 | 6 | |
| Anatomical score (points) | 80.3±6.0 | 80.7±5.4 | 0.903(P>0.05) |

Group A (mean 22.9±8.9%) versus Group B (61.5±15.7%), a 62.8% difference. These data indicate the multi-angle laser device (MLD) significantly improved efficiency and accuracy (all p<0.05).

Table 2 and Fig 5 summarize the imaging time, number of shots, and initial deviation between groups across views. For imaging time, Group A required less time than Group B for both AP and LAT in normal (57.4%, 38.0% decrease) and rotated positions (48.4%, 37.4% decrease). Similarly, Group A demonstrated fewer shots than Group B for AP and LAT in both positions (61.3%, 51.5% and 55.8%, 40.5% decrease). Regarding initial deviation, Group A showed markedly lower values than Group B across all views, ranging from 54.3% to 71.5% smaller in normal position and 59.5% to 65.1% smaller in rotated position. Together, this signifies the MLD reduced radiation and risks of complications by improving efficiency and accuracy for novice operators.

Fig 6 illustrates the per-shot deviations for both groups. Group A data points clustered in the bottom left region within 30% deviation using fewer shots, reflecting more precise and consistent imaging. Group B showed wider scattering exceeding 80% deviation with more variable adjustments. While Group A maintained higher accuracy throughout, the laser assistance benefit diminished during fine-tuning. Overall, the MLD improved initial positioning precision and minimized shots compared to traditional methods.

**Table 2. Comparison of imaging performance between groups.**

| Items | Group A(n = 20) | Group B(n = 20) | P-value |
|---|---|---|---|
| **Time of Shots (min)** | | | |
| AP(N) | 2.91±0.96 | 6.84±3.10 | 0.000006* |
| Lat(N) | 3.04±1.85 | 4.90±3.07 | 0.027* |
| AP(CR) | 4.63±3.25 | 8.97±3.84[c] | 0.001* |
| Lat(CR) | 3.08±1.36 | 4.92±2.05[b] | 0.005* |
| Total time | 13.66±5.61 | 25.63±8.17 | 0.00003* |
| **Number of Shots** | | | |
| AP(N) | 3.75±2.31 | 9.70±4.46 | 0.000006* |
| Lat(N) | 3.30±2.68 | 6.80±3.12[a] | 0.0002* |
| AP(CR) | 4.40±3.17 | 9.95±3.78 | 0.00003* |
| Lat(CR) | 3.60±2.35 | 6.05±2.14[b] | 0.0004* |
| Total Shots | 15.05±7.84 | 32.50±7.24 | 0.000001* |
| **Initial Shot Deviation (%)** | | | |
| AP(N) | 29.4±25.2 | 64.4±23.2 | 0.0002* |
| Lat(N) | 17.5±9.8 | 61.4±23.4 | 0.0000004* |
| AP(CR) | 24.3±15.8 | 69.7±24.8 | 0.000003* |
| Lat(CR) | 20.5±12.6 | 50.6±27.3[b] | 0.0001* |
| Mean Deviation | 22.9±8.9 | 61.5±15.7 | 0.00000008* |

Note: Statistical significance is indicated by *p < 0.05. AP refers to the anteroposterior view, Lat refers to the lateral view, N represents the model at the normal position, and CR represents the model at the crown rotation position.

[a]. Comparison between AP(N) and Lat(N), p < 0.05, statistically significant.

[b]. Comparison between AP(CR) and Lat(CR), p < 0.05, statistically significant.

[c]. Comparison between AP(N) and AP(CR), p < 0.05, statistically significant.

[d]. Comparison between Lat(N) and Lat(CR), p < 0.05, statistically significant.

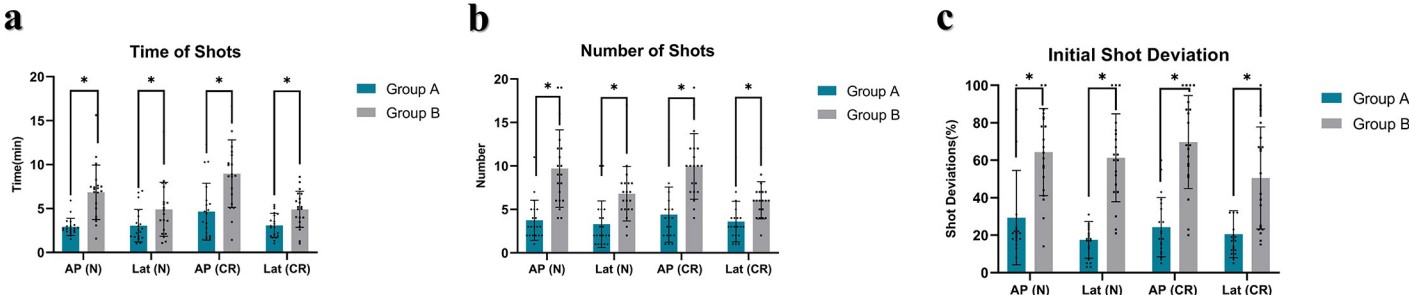

**Fig 5. Comparison of imaging parameters between groups.** (a) Time (min) (b) Shots (n) (c) Deviation (%). Note: * indicates p<0.05 between groups. AP—anteroposterior, Lat—lateral, N—normal position, CR—coronally rotated position.

### 3.3 Questionnaire results

Group A showed significantly higher scores than Group B in comfort, self-learning efficiency, and knowledge mastery (all p<0.05), but no difference in interest (p>0.05) (Table 3, Fig 7).

## 4 Discussion

### 4.1 C-arm fluoroscopy in surgery and education

C-arm fluoroscopy plays an indispensable role in minimally invasive spine surgery by enabling real-time imaging from different angles [8]. However, precisely maneuvering the mobile C-arm can be challenging for novice operators [9]. Their lack of experience often leads to improper positioning, prolonged radiation exposure, and compromised learning outcomes [10].

Incorporating technologies like the MLD into traditional C-arm education can assist novice operators in achieving efficient and accurate imaging [11]. The visual guidance provides real-time feedback to facilitate spatial conceptualization and minimize trial-and-error adjustments [12,13]. This enhances learning efficiency while reducing risks associated with excessive radiation and surgical complications [Wu, 2023 #3298].

### 4.2 C-arm laser navigation technology

C-arm laser navigation techniques have been increasingly developed to improve traditional fluoroscopic imaging [14]. By integrating laser devices with the C-arm, they can project surgical trajectories onto the patient's body to assist in positioning [11].

However, limitations exist including lower accuracy, unstable installation, and the need for recalibration [15]. Our MLD addresses these limitations with a simple and detachable structure, precise guidance without repeated debugging, and wide compatibility with most C-arms. This makes it suitable for broad clinical application.

Further optimization of C-arm laser navigation is warranted to maximize positioning accuracy while minimizing workload for operators. User-centered design considering ergonomics and workflow efficiency will be important. With technological improvements, C-arm laser devices could become standard equipment in operating rooms.

### 4.3 Effects of MLD on fluoroscopy performance

**4.3.1 MLD reduces fluoroscopy time, frequency, and initial deviation.** The MLD significantly reduced fluoroscopy time, frequency, and initial deviation compared to traditional

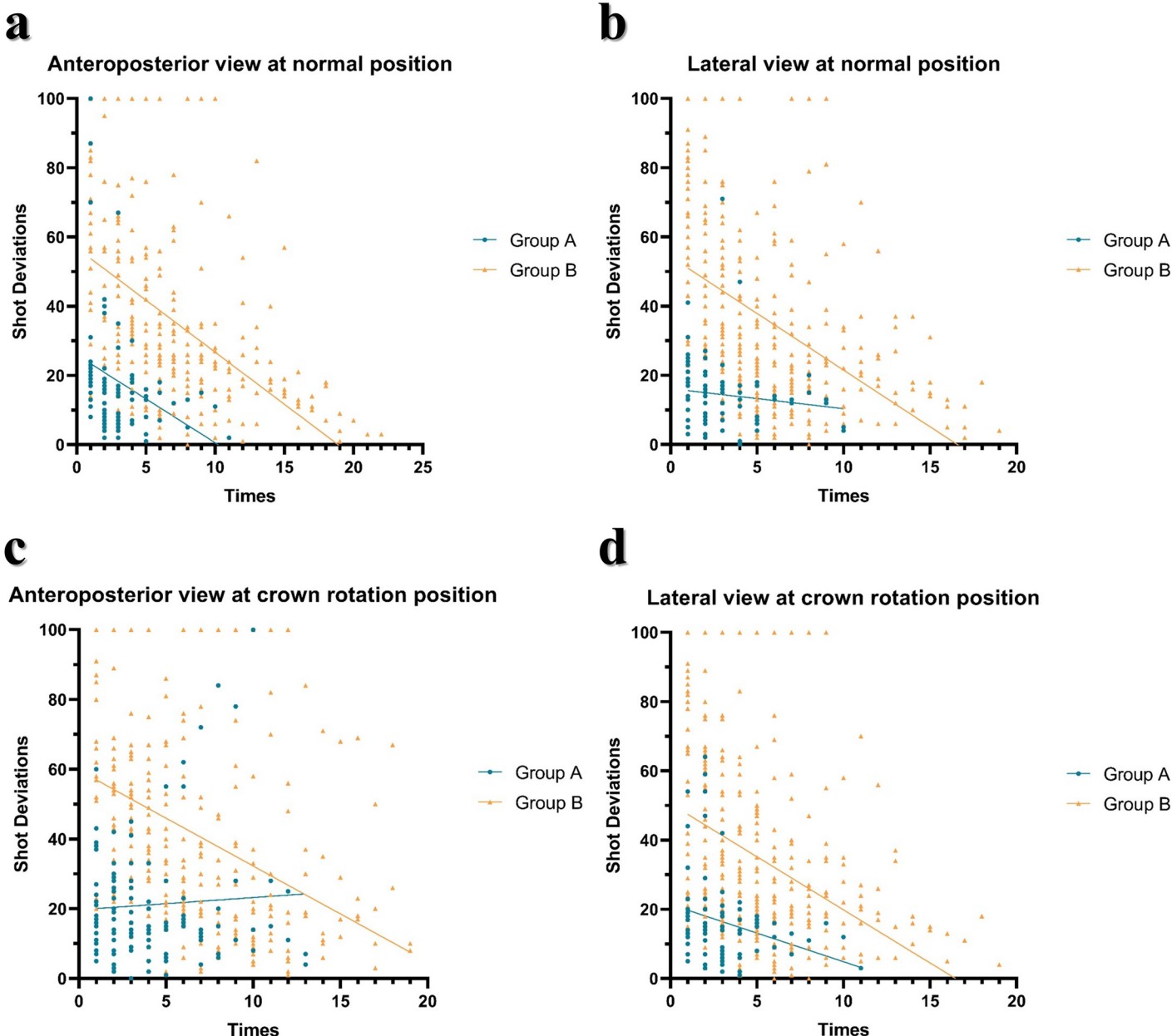

**Fig 6. Shot deviation plots by position and view.** AP, normal (b) Lateral, normal (c) AP, rotated (d) Lateral, rotated X-axis—shots; Y-axis—deviation. Note: Blue dots–Group A; Orange dots–Group B.

methods. This can be attributed to enhanced human-equipment interaction through visual guidance.

Fluoroscopy with C-arms is prone to human errors in positioning and adjustment, leading to poor images [16]. The MLD addresses equipment limitations that contribute to human errors by establishing spatial correspondence between lasers, X-rays, and anatomy [17]. This facilitates rapid and precise localization, aligning the X-ray to the region of interest.

**Table 3. Questionnaire ratings between groups.**

| Item (points) | Group A(n = 20) | Group B(n = 20) | P-value |
|---|---|---|---|
| Comfort | 4.15±0.93 | 2.65±0.81 | 0.00003* |
| Interest | 3.40±1.14 | 3.50±1.15 | 0.789 |
| Efficiency | 4.10±0.97 | 2.85±0.99 | 0.0002* |
| Mastery | 3.85±1.09 | 2.85±0.93 | 0.003* |
| Total score | 15.50±2.88 | 11.85±2.03 | 0.0002* |

Note:

*p < 0.05, statistically significant.

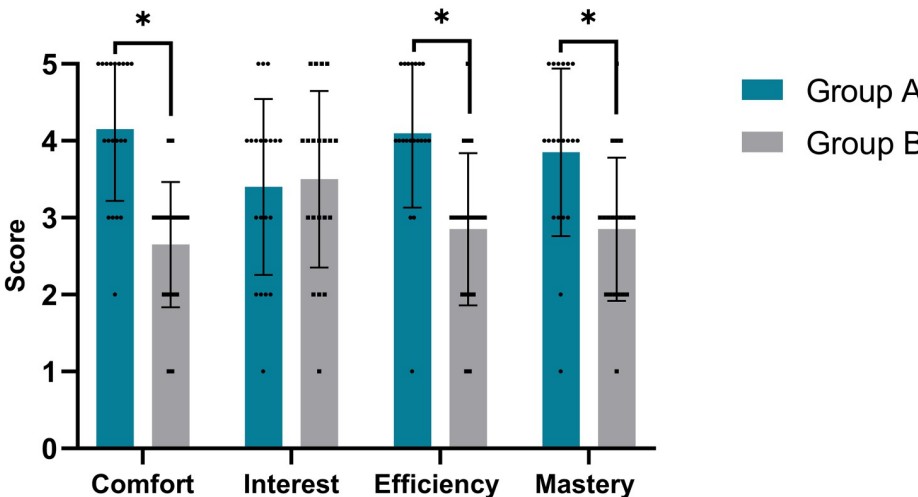

**Fig 7. Questionnaire scores between groups.** Note: * indicates p < 0.05, statistically significant.

Therefore, the MLD demonstrates value in reducing effortful trial-and-error adjustments by novice operators. It also minimizes risks associated with excessive radiation exposure and surgical complications resulting from improper or prolonged fluoroscopy.

The MLD establishes a spatial correspondence between the lasers, X-rays, and anatomy by strategically integrating perpendicular and angled laser beams with the C-arm. This allows the laser projections to visualize the underlying X-ray trajectories on the body surface. By enabling users to align the laser with the target position on the screen, the MLD provides visual guidance for rapid and accurate positioning [12,13].

Essentially, the MLD improves human-machine interaction by reducing trial-and-error adjustments and enhancing spatial orientation [11,16]. It assists novice operators in achieving precise and efficient fluoroscopy, minimizing radiation exposure. The simple and detachable structure also makes the MLD suitable for widespread clinical application.

**4.3.2 MLD improves initial positioning accuracy.** The MLD significantly reduced the initial deviation compared to traditional methods, indicating enhanced accuracy in initial positioning. However, the benefit diminished during subsequent smaller-scale adjustments.

The residual errors may be attributed to C-arm limitations in fine movements and the MLD's wide laser beam reducing localization precision [16]. As the target area narrows, the

laser guidance becomes less effective, requiring some trial-and-error similar to traditional methods.

This suggests potential value in upgrading the MLD with adjustable beam width or integrating with motorized C-arms for automated incremental adjustments. Further refinements could maximize accuracy across the full positioning workflow.

**4.3.3 MLD reduces the gap between AP and LAT performance.**   The traditional group showed longer time and more attempts for AP versus LAT images, likely due to C-arm limitations in anterior-posterior movements. However, the MLD group showed no significant AP versus LAT difference, indicating that visual guidance improved efficiency for AP positioning.

The perpendicular lasers assisted users in estimating required anterior-posterior adjustments for AP imaging. By providing real-time movement feedback, the MLD minimized the gap in performance between AP and LAT views compared to traditional methods.

This demonstrates the value of the MLD in overcoming C-arm maneuvering difficulties faced by novices to achieve uniformly efficient and accurate fluoroscopy across views [14].

## 4.4 Learner satisfaction with MLD

The MLD group reported higher scores for comfort, efficiency, and knowledge mastery compared to traditional methods. The visual guidance facilitated spatial orientation and provided positive feedback during positioning [14,16]. This increased learner confidence and simplified the acquisition process.

Both groups showed equally high interest, likely as a novel educational experience. While traditional training requires extensive hands-on practice [10], MLD accelerated learning by reducing technical barriers through real-time guidance [12,13].

Overall, the study demonstrates the value of the MLD for enhancing novice education. The ability to quickly obtain accurate images despite limited experience highlights the MLD's potential in surgical training programs. Further validation using larger clinical trainee samples would support integration into radiology curricula.

## 4.5 Limitations

This study has several limitations. First, the C-arm was not reset between AP and LAT imaging. Second, axial and sagittal rotations were not evaluated. Third, the phantom did not simulate deformities or scoliosis. Fourth, the specific effects of the 45˚ and 135˚ lasers warrant further investigation. Finally, the small student sample from one university may limit generalizability.

Future studies should utilize larger clinical trainee samples, optimize experimental procedures, and evaluate applications of specific laser angles. This will further validate the educational value of the MLD and support integration into radiology training programs.

## 5 Conclusion

This study demonstrates the educational value of the multi-angle laser-assisted device for improving novice learning of C-arm fluoroscopy in simulated lumbar spine surgery. Compared to traditional methods, the MLD significantly reduced imaging time, number of attempts, and initial deviation while enhancing learner satisfaction. By providing real-time visual guidance, the MLD facilitated efficient and accurate positioning despite limited experience. These findings support the integration of the MLD into radiology training curricula as an effective educational tool to assist novice operators in achieving precise and efficient surgical localization. Further clinical validation studies are warranted to optimize MLD applications for maximizing novice performance.

## Supporting information

**S1 Data.**
(XLSX)

## Author Contributions

**Conceptualization:** Chun-Mei Chen.

**Data curation:** Yuan-Dong Zhuang, Rui-Jin Li, Jia-Jun Wu, Xue-Wei He, Wen-Bin Zou, Xu-Chu Xu, Si-Qi Lu.

**Formal analysis:** Yuan-Dong Zhuang, Rui-Jin Li, Jia-Jun Wu, Xue-Wei He, Si-Qi Lu.

**Funding acquisition:** Yuan-Dong Zhuang, Chun-Mei Chen.

**Investigation:** Yuan-Dong Zhuang, Rui-Jin Li, Xue-Wei He, Wen-Bin Zou, Xu-Chu Xu, Si-Qi Lu.

**Methodology:** Yuan-Dong Zhuang, Rui-Jin Li, Jia-Jun Wu, Xue-Wei He, Wen-Bin Zou, Si-Qi Lu, Chun-Mei Chen.

**Visualization:** Rui-Jin Li.

**Writing – original draft:** Yuan-Dong Zhuang, Rui-Jin Li.

**Writing – review & editing:** Yuan-Dong Zhuang.

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
