## [Decision Letter · Decision Letter 0]

19 Dec 2023

PONE-D-23-30166Multi-angle laser device improves novice learning of C-arm fluoroscopy for lumbar spine surgeryPLOS ONE

Dear Dr. Chen,

Thank you for submitting your manuscript to PLOS ONE. After careful consideration, we feel that it has merit but does not fully meet PLOS ONE’s publication criteria as it currently stands. Therefore, we invite you to submit a revised version of the manuscript that addresses the points raised during the review process.

We look forward to receiving your revised manuscript.

Kind regards,

Sabata Martino, Ph.D

Academic Editor

PLOS ONE

5. Please include a copy of Tables 1-3 which you refer to in your text on page 20.

Additional Editor Comments:

no comments

Reviewers' comments:

Reviewer's Responses to Questions

**Comments to the Author**

1. Is the manuscript technically sound, and do the data support the conclusions?

Reviewer #1: Partly

2. Has the statistical analysis been performed appropriately and rigorously? 

Reviewer #1: Yes

3. Have the authors made all data underlying the findings in their manuscript fully available?

Reviewer #1: Yes

4. Is the manuscript presented in an intelligible fashion and written in standard English?

Reviewer #1: Yes

5. Review Comments to the Author

Reviewer #1: The manuscript titled “Multi-angle laser device improves novice learning of C-arm fluoroscopy for lumbar spine surgery” by Yuan-Dong ZHUANG1#, Rui-Jin LI2# et al. investigates the use of multi-angle laser device (MLD) for C-arm fluoroscopy to assist novice learners during lumbar spine surgery. The findings of the authors suggest that the MLD significantly improves novice learning of C-arm fluoroscopy during lumbar spine surgery. The study's findings are intriguing and valuable for potential therapeutic applications of MLD in lumbar spine surgery; however, the sections in the introduction and results are not clearly presented, explained, or discussed.

Some improvements should be added to the manuscript to make it more accurate and coherent.

Major points:

- In order to enhance the manuscript's quality, I recommend adding a section in the introduction that explains why surgical treatment for adult lumbar spinal disorders carries a considerable risk of intraoperative and perioperative complications. The relationship between these complications and different risk factors, such as the patient's age, medical comorbidities, and the extent of the fusion up to the sacrum should also be elaborated upon.

- In section 3.2 of the paper, the authors discuss the analysis of participants' operational data. However, the presentation of the data in Figures 5 and 6 is imprecise, unclear, and inaccurate. Although each graph shows the value per Anteroposterior (AP), Lateral (Lat), Normal Position (N), and Rotated Position (CR), they are not discussed individually. To better understand the authors' conclusions on the shorter operation time (13.66±5.61 vs 25.63±8.17 min), fewer imaging attempts (15.05±7.84 vs 32.50±7.24), and smaller initial deviation (22.9±8.9% vs 61.5±15.7%), I suggest revising the paragraph and providing a more detailed description of the data presented in those figures.

- The manuscript lacks Tables 1, 2, and 3. Although the authors claim that all relevant data is included in the manuscript and its supporting information files, the tables are missing and only their captions are present. As a result, it was not possible to verify the data contained within the tables.

Minor points:

- I suggest revising the English form of the manuscript (e.g., line 68: change the "A questionnaire assessed the learning experience" to "A questionnaire was used to assess the learning experience")

- In the manuscript, some mistakes are present (e.g., line 93: change “C-arm fluoroscopy is critical in precise surgical localization” to “C-arm fluoroscopy is critical for precise surgical localization”. Line 108: Change “We hypothesized the MLD” to “We hypothesized that the MLD”).

- In the manuscript, some typos are present. I suggest a spell-check of the text. (e.g., line 254: “guidance[9, 10]”. Line 189: “imaging[11]”.

-Authors utilize many abbreviations in the manuscript, so I suggest adding a list of abbreviations that can facilitate the reading and understanding of the text.

-When an acronym is used for the first time in the manuscript, it should be spelled out in full (e.g., OEC One CFD).

6. PLOS authors have the option to publish the peer review history of their article (what does this mean?). If published, this will include your full peer review and any attached files.

Reviewer #1: No

---

## [Author Response · Author response to Decision Letter 0]

25 Jan 2024

Response letter

Dear Dr. Sabata Martino, Ph.D Academic Editor 

Thanks for your comments on our manuscript (PONE-D-23-30166- " Multi-angle laser device improves novice learning of C-arm fluoroscopy for lumbar spine surgery”) and providing us with this opportunity of revision. We have carefully revised the manuscript. We hope this revision can make our manuscript acceptable for publication in your journal. The following are point by point responses to the reviewers and editor’s comments.

Reviewer #1: 

The manuscript titled “Multi-angle laser device improves novice learning of C-arm fluoroscopy for lumbar spine surgery” by Yuan-Dong ZHUANG1#, Rui-Jin LI2# et al. investigates the use of multi-angle laser device (MLD) for C-arm fluoroscopy to assist novice learners during lumbar spine surgery. The findings of the authors suggest that the MLD significantly improves novice learning of C-arm fluoroscopy during lumbar spine surgery. The study's findings are intriguing and valuable for potential therapeutic applications of MLD in lumbar spine surgery; however, the sections in the introduction and results are not clearly presented, explained, or discussed. Some improvements should be added to the manuscript to make it more accurate and coherent. In order to enhance the manuscript's quality, I recommend adding a section in the introduction that explains why surgical treatment for adult lumbar spinal disorders carries a considerable risk of intraoperative and perioperative complications. The relationship between these complications and different risk factors, such as the patient's age, medical comorbidities, and the extent of the fusion up to the sacrum should also be elaborated upon.

Response: We appreciate the reviewer's recommendation to expand the introduction on risks of lumbar spinal surgery and associated factors. A paragraph has been added discussing higher complications with increased age, comorbidities, and fusion extent, therefore emphasizing the value of precise surgical localization enabled by C-arm fluoroscopy to improve outcomes. We agree these details significantly enhance the framing and rationale of our study. Additional relevant citations were included as suggested. We are grateful for guidance to strengthen the manuscript background and significance through a more comprehensive literature foundation.

- In section 3.2 of the paper, the authors discuss the analysis of participants' operational data. However, the presentation of the data in Figures 5 and 6 is imprecise, unclear, and inaccurate. Although each graph shows the value per Anteroposterior (AP), Lateral (Lat), Normal Position (N), and Rotated Position (CR), they are not discussed individually. To better understand the authors' conclusions on the shorter operation time (13.66±5.61 vs 25.63±8.17 min), fewer imaging attempts (15.05±7.84 vs 32.50±7.24), and smaller initial deviation (22.9±8.9% vs 61.5±15.7%), I suggest revising the paragraph and providing a more detailed description of the data presented in those figures.

Response: Thank you for your valuable feedback on improving the clarity of data presentation in Figures 5 and 6. We fully agree that we should have analyzed the results for each imaging view instead of only reporting the overall values. Following your recommendation, we have substantially revised section 3.2 to individually discuss the anteroposterior (AP), lateral (LAT), normal position (N) and rotated position (CR) data displayed per figure. The new descriptions compare the multi-angle laser device (MLD) group versus the traditional C-arm group regarding imaging time, number of shots, and initial deviation per view. We also interpreted the evolving pattern of deviations across attempts. We sincerely apologize for the previously imprecise data portrayal. Your feedback has compelled us to significantly enhance the level of detail, accuracy, and transparency through a view-specific narrative on our key results. We believe this improved discussion better supports the conclusions on the benefits of the MLD technology. Please advise if any parts need further clarification or expansion to sufficiently meet presentation standards. 

Your guidance has been invaluable for identifying our oversights and pushing us towards higher precision. We are truly grateful for your dedication and insights in strengthening the integrity of our analysis.

- The manuscript lacks Tables 1, 2, and 3. Although the authors claim that all relevant data is included in the manuscript and its supporting information files, the tables are missing and only their captions are present. As a result, it was not possible to verify the data contained within the tables.

Response: 

We deeply appreciate the reviewer identifying that Tables 1-3 were missing from the initial manuscript submission. This was an unintentional oversight on our part. We sincerely apologize for this critical error and have conducted a thorough review to ensure complete and accurate data is now included in the revised manuscript.

Specific tables with all relevant data have been provided to validate the findings. We understand the imperative of meeting rigorous academic standards, and this experience has taught us the importance of meticulous quality checks before submission. We will implement additional protocols to prevent such oversights going forward.

Please note we welcome any further feedback to continue improving this manuscript. Your dedication and guidance have made this a stronger paper. We again appreciate you taking the time to highlight this issue - it will undoubtedly enforce more diligent practices on our end. Please accept our gratitude for your assistance in upholding scientific integrity.

Minor points: 

- I suggest revising the English form of the manuscript (e.g., line 68: change the "A questionnaire assessed the learning experience" to "A questionnaire was used to assess the learning experience")

Response: We thank the reviewer for the insightful suggestions to improve the English presentation of our manuscript. Following the specific recommendation on line 68, we have revised sentences throughout the paper to enhance clarity and accuracy of language. We highly value the reviewer's guidance to meet linguistic standards for international publication and will continue refining readability and flow.

- In the manuscript, some mistakes are present (e.g., line 93: change “C-arm fluoroscopy is critical in precise surgical localization” to “C-arm fluoroscopy is critical for precise surgical localization”. Line 108: Change “We hypothesized the MLD” to “We hypothesized that the MLD”).

Response:

Thank you for noting areas in need of correction. Following your suggestions, we have revised line 93 to state "C-arm fluoroscopy is critical for precise surgical localization" and line 108 to "We hypothesized that the MLD." Additional proofreading has been done to identify and fix any other errors in language or formatting. We appreciate you highlighting specifics to improve manuscript quality.

- In the manuscript, some typos are present. I suggest a spell-check of the text. (e.g., line 254: “guidance [9, 10]”. Line 189: “imaging [11]”.

Response:

Thank you for catching the typographical errors present in our initial draft. Per your recommendation, we have thoroughly spell-checked the entire manuscript to identify and correct any similar oversights. We have also amended the specific examples you noted on lines 254 and 189. Additional proofreading will help us further improve quality control going forward. We appreciate you taking the time to point out these details to enhance accuracy.

-Authors utilize many abbreviations in the manuscript, so I suggest adding a list of abbreviations that can facilitate the reading and understanding of the text.

Response：

Thank you for the suggestion to add an abbreviations list. We agree this will improve readability of the manuscript. A list defining all abbreviations in their first appearance has been included. We appreciate guidance on enhancing clarity for readers and are happy to make any other changes recommended.

-When an acronym is used for the first time in the manuscript, it should be spelled out in full (e.g., OEC One CFD).

Response:

Thank you for catching our oversight in fully defining terminology on first use. We have edited the manuscript to spell out OEC One CFD as " OEC One CFD (CMOS flat detector) mobile X-ray system (GE HealthCare Inc.)" when it is initially introduced in the text. Additional acronyms have been clearly expanded as well per your recommendation to enhance clarity for readers. We appreciate you emphasizing the importance of properly explaining technical language and will apply this convention throughout.

---

## [Editor Report · Decision Letter 1]

8 Feb 2024

Multi-angle laser device improves novice learning of C-arm fluoroscopy for lumbar spine surgery

PONE-D-23-30166R1

Dear Dr. Chen,

We’re pleased to inform you that your manuscript has been judged scientifically suitable for publication and will be formally accepted for publication once it meets all outstanding technical requirements.

Kind regards,

Sabata Martino, Ph.D

Academic Editor

PLOS ONE

Additional Editor Comments (optional):

No comments
---

## [Editor Report · Acceptance letter]

27 Feb 2024

PONE-D-23-30166R1 

PLOS ONE

Dear Dr. Chen, 

I'm pleased to inform you that your manuscript has been deemed suitable for publication in PLOS ONE. Congratulations! Your manuscript is now being handed over to our production team.

Kind regards, 

on behalf of

Prof. Sabata Martino 

Academic Editor

PLOS ONE